



# Improve Ocean Modelling Software NEMO 4.0 benchmarking and communication efficiency

Gaston Irrmann [1,3], Sébastien Masson [1], Éric Maisonnave [2], David Guibert [3], and Erwan Raffin [3]

[1]LOCEAN-IPSL, Sorbonne Universités (UPMC)/IRD/CNRS/MNHN, UMR7159, Paris, France
[2]CERFACS/CNRS, CECI, Toulouse, France
[3]CEPP - Center for Excellence in Performance Programming, Atos Bull, 35700 Rennes, France

**Correspondence:** Gaston Irrmann (gaston.irrmann@locean-ipsl.upmc.fr)

**Abstract.** Communications in distributed memory supercomputers are still limiting scalability of geophysical models. Considering the recent trends of the semiconductor industry, we think this problem is here to stay. We present the optimisations that have been implemented in the actual 4.0 reference version of the ocean model NEMO 4.0 to improve its scalability. Thanks to the collaboration of oceanographers and HPC experts, we identified and removed the unnecessary communications in two

bottleneck routines, the computation of free surface pressure gradient and the forcing in the straights or unstructured open boundaries. Since a wrong parallel decomposition choice could undermine computing performance, we impose its automatic definition in all cases, including when subdomains containing land points only are excluded from the decomposition. For a smaller audience of developers and vendors, we propose a new benchmark configuration, easy to use while offering the full complexity of operational versions.

## 10    1    Introduction

There is, hopefully, no more need to justify the importance of climate research for our societies (Masson-Delmotte et al., 2018). Climate studies explore a complex system driven by a large variety of interactions between physical and bio-geo-chemical processes over ocean, atmosphere, land surface and cryosphere. Numerical modeling is an essential tool in climate research, which supplements the sparse observations in time and space. Numerical experiments are a unique way to tests hypotheses, to investigate which processes are at stake, to quantifying their impacts on the climate and its variability and, last

but not lest, to perform climate change forecasts (Flato et al., 2013). The numerical performance of climate models is key and must be kept at the best possible level in order to minimize the time-to-solution but also the energy-to-solution. Models must continuously evolve in order to take advantage of new machines. Exascale is expected in the coming years (the supercomputer Fugaku, ranked number 1 of November 2020 TOP 500 list, achieved a Linpack performance of 0.44 EFlop/s) but the growing

complexity of the supercomputers makes it harder and harder for model developers to catch-up with the expected performance.

Scalability is a major issue as models will have to achieve good scaling performance on hundred of thousands or even millions of a mix of tasks and threads (Etiemble, 2018). In addition to the hardware constrain, the community interest in better representing fine spatial scale phenomena pushes for finer and finer spatial resolution which can only be achieved through an





increase of the parallel decomposition of the problem. The costly communications between parallel processes must thus be
minimised in order to keep the time restitution of the numerical experiments at its best level.

In that perspective, this document focuses on the "NEMO" model (Nucleus for European Modelling of the Ocean, http://www.nemo-ocean.eu, Madec and Team), a framework for research activities and forecasting services in ocean and climate sciences, developed in a sustainable way by a European consortium. The constant renewal of the model equations on one hand and the evolution of the supercomputers technology on the other hand require that model computing performance must be continually
reviewed and improved. It is a delicate work to optimise a model like NEMO, which is used by a large community which profiles are ranging from Ph.D. students to experts in climate physics and modelling or operational oceanography. This work must improve the performance while preserving the code accessibility by climate scientists who use it and develop it, and are not necessary experts in computing sciences. This optimization work lies within this framework and we gathered, in this study, authors with very complementary profiles: oceanographers, NEMO developers, specialized engineers in climate modelling
and frontier simulations and pure HPC engineers. This works complements the report of Maisonnave and Masson (2019) by presenting the new HPC optimisations that have been implemented in NEMO 4.0, the actual reference version of the code.

Whether the core number increase in the future machines will affect the number of cores per share memory node, the total machine node number, or both, is unclear. In NEMO, parallel subdomains are not sharing memory and the MPI library is required to exchange the model variables at their boundaries. This is why the incremental approach we follow is addressing
and facilitating the future addressing of the problem of inter and intra node MPI communication cost. We assume that its reduction is and will stay worthwhile, independently of hardware evolution.

We first describe the new features that have been added to the code in order to support the optimisation work, see Sect. 2. This includes an automatic definition of the domain decomposition (Sect. 2.1), which minimizes the subdomains size for a given maximum number of cores and takes into account the possibility to remove subdomains containing only land points. We
next present, in Sect. 2.2, the new benchmark test case that was specifically designed to be extremely simple to be deployed while being able to represent all physical options and configurations of NEMO. The optimisation work by itself is detailed in Sect. 3. A significant reduction of the number of communications is first proposed in the computation of free surface pressure gradient (Sect. 3.1). The second set of optimisation concerns the communications in the handling of the straits or unstructured open boundaries, Sect. 3.2. The last section (Sect. 4) discusses and concludes this work.

## 2   A proper benchmarking environment


This second section of the paper details the code modifications introduced in NEMO 4.0 in order to provide a proper benchmarking environment aiming at facilitating numerical performance tests and optimizations.

### 2.1   Optimum Dynamic Sub-Domain Decomposition

The MPI implementation in NEMO uses a horizontal domain decomposition: the first 2 dimensions are divided by 2 coefficients
(called $jpni$ and $jpnj$) allowing to distribute the MPI tasks over $jpni \times jpnj$ rectangular subdomains (see chapter 8.3 in Madec





and Team). Note that the land only subdomains can be suppressed from the computational domain. The number of MPI task is, in this case, smaller than $jpni \times jpnj$ (see figure 8.6 in Madec and Team).

The choice of the domain decomposition proposed by default in NEMO until version 3.6 was very basic as 2 was the only prime factor considered when looking at divisors of the number of MPI tasks! Tintó et al. (2017) underlined this deficiency.
In their figure 4, they demonstrated that the choice of the domain decomposition is a key factor to get the appropriate model scalability. In their test with ORCA025 configuration on 4096 cores, the number of simulated years per day is almost multiplied by a factor of 2 when using their optimum domain decomposition instead of the default one. Benchmarking NEMO with the default domain decomposition would therefore be completely misleading. Tintó et al. (2017) proposed a strategy to optimize the choice of the domain decomposition in a preprocessing phase. Finding the best domain decomposition is thus the starting
point of any work dedicated to NEMO numerical performance.

We detail in this section how we implemented a similar approach, but on-line in the initialization phase of NEMO. Our idea is to propose, by default, the best domain decomposition for a given number of MPI tasks and avoid the waste of cpu resources by non-expert users.

### 2.1.1 Optimal domain decomposition research algorithm

Our method is based on the minimization of the size of the MPI subdomains taking into account the fact that land only subdomains can be excluded from the computation. The algorithm we wrote can be summarized in 3 steps:

1. Get the Land Fraction ($LF$: the total number of land points divided by the total number of points, thus between 0 and 1) of the configuration we are running. The Land Fraction will provide the maximum number of subdomains that could be potentially removed from the computational domain. If we want to run the model on $Nmpi$ processes we must look for
a domain decomposition generating a maximum of $Nsub_{max} = \lfloor Nmpi/(1 - LF) \rfloor$ subdomains, as we wont be able to remove more than $Nmpi \times LF$ land only subdomains.

2. We next have to provide the best domain decomposition defined by the following rules: (a) it generates a maximum of $Nsub_{max}$ subdomains, (b) it gives the smallest subdomain size for a given value $Nsub$ of subdomains and (c) no other domain decomposition with less subdomains has a subdomain size smaller or equal. This last constrain requires in fact
that we build the list of best domain decompositions incrementally, from $Nsub = 1$ to $Nsub = Nsub_{max}$.

3. Having this list, we have to test the largest value of $Nsub$ that is listed and check if once we remove its land only subdomains, we obtain a number of remaining ocean subdomains lower or equal than $Nmpi$. If it is not the case, we discard this choice and test the next domain decomposition listed (in a decreasing order of number of subdomains) until it fits the limit of $Nmpi$ processes.

Note that we could imagine that in a few cases, a non-optimal domain decomposition would allow to remove more land only subdomains than the selected optimal decomposition and become a better choice. Taking into account this possibility would increase tenfold the number of domain decompositions to be tested, which would make the selection extremely costly





and impossible to use. We consider that the probability of facing such a case becomes extremely unlikely as we increase the number of subdomains (which is the usual target) as smaller subdomains fit better the coastline. We therefore ignore this

possibility and consider only optimal decomposition when looking at land only subdomains.

If the optimal decomposition found has a number of ocean subdomains $Nsub$ smaller than $Nmpi$, we print a warning message saying that the user provides more MPI tasks than needed and we simply keep in the computational domain ($Nmpi -$ $Nsub$) land only subdomains in order to fit the required number of $Nmpi$ MPI tasks. This simple trick that may look quite useless for simple configurations is in fact required when using AGRIF (Debreu et al., 2008) because (as it is implemented

today in NEMO) each parent and child domains share the same number of MPI tasks that can therefore rarely be optimized to each domain at the same time.

The next 2 sub-sections details the keys parts of steps (1,3) and (2).

### 2.1.2 Getting land-sea mask information

We need to read the global land-sea information for 2 purposes: get the land fraction (step 1) and find the number of land

only subdomains in a given domain decomposition (step 3). By default, reading NetCDF configuration files in NEMO can be trivially done by the mean of a dedicated Fortran module ("iom"). However, in this specific context, things must be done more carefully to avoid a one-off but potentially extremely large memory allocation associated with a massive disk access. Until revision 3.6 included, each MPI process was reading the whole global land-sea mask, which is clearly not the proper strategy when aiming at running very large domains on large numbers of MPI processes with limited memory. The difficulty here lies

in the minimization of the memory used to get the needed information from the global land-sea mask. To overcome this issue, we dedicate some processes to read only horizontal stripes of the land-sea mask file. This solution requires less memory and will ensure the continuity of the data to be read which optimizes the reading process.

When looking at the land fraction for step (1), we need the total number of ocean points, and we use as much processes as we have to read the file with 2 limits: each process must read at least 2 lines and we use no more that 100 processes to avoid to

saturate the file system with too many processes accessing the same input file (an arbitrary value that could be changed). The number of oceanic points in each stripe is next added and shared among processes through a collective communication.

When looking at the land only subdomains in a $jpni \times jpnj$ domain decomposition for step (3), we need the number of ocean points in each of the $jpni \times jpnj$ subdomains. In this case, we read $jpnj$ stripes of land-sea mask corresponding to bands of subdomains. This work is distributed among a maximum of the $jpnj$ processes accessing the input file concurrently.

Each of these processes reads sequentially one or several stripes of data and communicates only the number of ocean points for the $jpni$ subdomains included in data stripes loaded in memory.

### 2.1.3 Getting the best domain decomposition sorted from 1 to $Nsub_{max}$ subdomains

The second step of our algorithm starts with the simple fact that domain decomposition uses Euclidean division: the division of the horizontal domain by the number of processes along i and j directions rarely results in whole numbers. In consequence,

some MPI subdomains will potentially be 1-point larger in i and/or j direction than others. Increasing the number of processes





does not always reduce the size of the largest MPI subdomains, especially when using a large number of processes compared to the global domain size. Table 1 illustrates this point with a simple example: a 1D domain of 10 points ($jpiglo$) distributed among $jpni$ tasks with $jpni$ ranging from 1 to 9. Because of the halos required for MPI communications, the total domain size is ($jpiglo + 2 * jpni - 2$) that must be divided by $jpni$. One can see that using $jpni = 4$ to 7 will always provide the same size for the largest subdomains: 4. Only specific values of $jpni$ (1, 2, 3, 4 and 8) will end up in a reduction of the largest subdomains size and correspond to the optimized values of $jpni$ that should be chosen. Using other values would simply increase the number of MPI subdomains that are smaller without affecting the size of the largest subdomain.

| $jpiglo$ | 10 | | | | | | | | |
|---|---|---|---|---|---|---|---|---|---|
| $jpni$ | 1 | 2 | 3 | 4 | 5 | 6 | 7 | 8 | 9 |
| $jpiglo + 2jpni - 2$ | 10 | 12 | 14 | 16 | 18 | 20 | 22 | 24 | 26 |
| $(jpiglo + 2jpni - 2)/jpni$ | 10 | 6.0 | 4.66 | 4.0 | 3.6 | 3.33 | 3.14 | 3.0 | 2.88 |
| $jpi_{max}$ | 10 | 6 | 5 | 4 | 4 | 4 | 4 | 3 | 3 |

**Table 1.** Example of 1D 10-point domain decomposition

This result must be extended to the 2D domain decomposition used in NEMO. When searching all couples ($jpni, jpnj$) that should be considered when looking for the optimal decomposition, we can quickly reduce their number by selecting $jpni$ and $jpnj$ only among the values suitable for the 1D decomposition of $jpiglo$ along the i direction and $jpjglo$ along the j direction. The number of these values corresponds roughly to the number of divisor of $jpiglo$ and $jpjglo$, which can be approximated by $2\sqrt{jpiglo}$ and $2\sqrt{jpjglo}$. This first selection is thus providing about $2\sqrt{jpiglo \times jpjglo}$ couples ($jpni, jpnj$) instead of the ($jpiglo \times jpjglo$) couples that could be defined by default. Next, we discard from the list of couples ($jpni, jpnj$) all cases ending up with more subdomains than $Nsub_{max}$ provided by the previous step.

The final part of this second step is to build the list of optimal decompositions, each one defined by a couple ($jpni, jpnj$), with a number of subdomains ($jpni \times jpnj$) ranging from 1 to a maximum of $Nsub_{max}$. This work is done with an iterative algorithm starting with the couple ($jpni, jpnj$) = (1, 1). The recurrence relation to find element $N + 1$ knowing element $N$ of the list of optimal decompositions is the following: first, we keep only the couples ($jpni, jpnj$) which maximum subdomain size is smaller than the maximum subdomain size found for the element $N$. Next, we define the element $N + 1$ as the couple ($jpni, jpnj$) that gives the smallest number of subdomains ($jpni \times jpnj$). It happens rarely that several couples ($jpni, jpnj$) correspond to this definition. In this case, we decided to keep the couple with the smaller subdomain perimeter, so the volume of data exchanged during the communications is minimized. This choice is quite arbitrary as NEMO scalability is limited by the number of communications but not by their volume. Limiting the subdomain perimeter should therefore have a very limited effect. We stop the iteration process once there is no more couple with a subdomain size smaller than the one selected at rank $N$.

Once this list of the best domain decomposition sorted from 1 to $Nsub_{max}$ subdomains is established, we just have to follow the third step of our algorithm to get the best subdomain decomposition (see section 2.1.1).



## 2.2 The BENCH configuration

Once we found the best domain decomposition, exploring the code numerical performance requires a proper benchmark that
is easy to install and configure but, at the same time, keeps the code usage close to the production mode.

The NEMO framework proposes various configurations based on a core ocean model: e.g. global (ORCA family) or regional grids, with different vertical and horizontal spatial resolution. Different components can moreover be added to the ocean dynamical core: sea-ice (i.e. SI3) and/or bio-geo-chemistry (i.e. TOP-PISCES). A comprehensive performance analysis must thus be able to scrutinise the subroutines of every module sets to deliver a relevant message about the computing performance of the
NEMO model to its whole users community. On the other hand, as pointed out by the former RAPS consortium (Mozdzynski, 2012), performing such benchmarking exercise must be kept simple, since it is often done by people with basic knowledge of NEMO or physical oceanography (e.g. HPC experts and hardware vendors).

We detail in this section a new NEMO configuration we specifically developed to simplify future benchmarking activities by responding to the double constraint of (1) be light and trivial to use and (2) allow to test any NEMO configuration (size,
periodicity pattern, components, physical parameterizations...) At the opposite of the dwarf concept (Müller et al., 2019), this configuration encompasses the full complexity of the model and helps to address the current issues of the community. This BENCH configuration was used in (Maisonnave and Masson, 2019) to assess the performances of the global configuration (ORCA family). We use it in the current study section to continue this work and improve now the performances of the regional configurations.

## 2.2.1 BENCH general description

This new configuration, called BENCH, is made available in the tests/BENCH directory of the NEMO distribution.

BENCH is based on a flat bottom rectangular cuboid, which allows to by-pass any input configuration file and gives the possibility to define the domain dimensions simply via namelist parameters ($nn\_isize$, $nn\_jsize$ and $nn\_ksize$ in $namusr\_def$). Note that the horizontal grid resolution is fixed (100km) whatever grid size is defined, which limits the growth of instabilities
and allows to keep the same time step length in all tests.

Initial conditions and forcing fields do not correspond to any reality and have been specifically designed (1) to ensure the the maximum of robustness and (2) to facilitate BENCH handling as they do not require any input file. In consequence, BENCH results are meaningless from a physical point of view and should be used only for benchmarking purposes. The model temperature and salinity are almost constant everywhere with a light stratification which keep the vertical stability of the
model. We add on each point of each horizontal levels a very small perturbation which keeps the solution very stable but lets the oceanic adjustment processes occurring and the associated amount of calculations at a realistic level. This perturbation also guarantees that each point of the domain has a unique initial value of temperature and salinity, which facilitates the detection of potential MPI issues. We apply a zero forcing at surface, except for the wind stress which is small and slowly spatially varying.

To make it as simplest as possible to use, the BENCH configuration does not need any input files, so that a full simulation can
be lead without any other download but the source code. In addition, the lack of input (or output) files prevents any disk access





perturbation of our performance measurement. However, output can be activated as in any NEMO simulation, for example to make possible a performance evaluation of the XIOS (Meurdesoif, 2018) output suite.

### 2.2.2 BENCH flexibility

Any NEMO numerical schemes and parameterizations can be used band be used in BENCH. To help the user in his namelist

choices and to test diverse applications, a selection of namelist parameters is provided with BENCH to confer to the benchmarks the numerical properties corresponding to popular global configurations: ORCA1, ORCA025 and ORCA12.

The SI3 sea-ice and TOP-PISCES modules can be activated or not by choosing the appropriate namelist parameters and CPP keys when compiling. The initial temperature/salinity definition was designed in such a was that if SI3 is activated in the simulation, the sea ice will cover about 1/5 of the domain, which corresponds approximately of the annual ratio of ORCA

ocean grid points covered by sea-ice.

Any closed or periodical conditions can be used in BENCH and specified through a namelist parameter ($nn\_perio$ in $namusr\_def$). User can for example, use closed boundaries, East-West periodical conditions, bi-periodic conditions ($nn\_perio$ = 7, to make sure that all MPI subdomains have exactly 4 neighbours) or even the north pole folding ($nn\_perio$ = 4 or 6) used in the global configuration ORCA with its bi-polar grid in the northern hemisphere. The specificity of the periodical conditions

in ORCA global grids have a big impact on performance, which motivated the possibility to use them in new BENCH configuration: a simple change of $nn\_perio$ definition gives to BENCH the ORCA periodic characteristics and reproduces the same communication pattern between subdomains located on the Northernmost part of the grid.

The previous section describes BENCH main features that are set by default but we must keep in mind that each of them can be modified through namelist parameters if needed. By default, BENCH is not using any input file to define the configuration,

the initial conditions and the forcing fields. We could however decide to specify some input files if it was requited by the functionality to be tested.

### 2.2.3 BENCH grid size, MPI domain decomposition and land only subdomains

Like in any other NEMO configuration, BENCH computations can be distributed on a set of MPI processes. BENCH grid size is defined in the namelist with 2 different options. If $nn\_isize$ and $nn\_jsize$ are positive, they simply define the total grid size

on which the domain decomposition will be applied. This is the usual case, the size of each MPI subdomains is comparable (but not necessary equal) for each MPI task and computed by the code according to chosen pattern of domain decomposition. If $nn\_isize$ and $nn\_jsize$ are negative the absolute value of these parameters will no more define the global domain size but the MPI subdomains size. In this case, the size of the global domain is computed by the code to fit the prescribed subdomains size. This options forces each MPI subdomains to have the exact same size whatever number of MPI processes is used, which

facilitates the measurement of model weak scalability.

In NEMO, calculations are processed, in a large majority, at each grid point and a mask is next applied to take into account land grid points, if any. Consequently, the amount of calculations is the same with or without bathymetry, and the computing performance of BENCH and a realistic configuration are extremely close. However, we must underline that the absence of



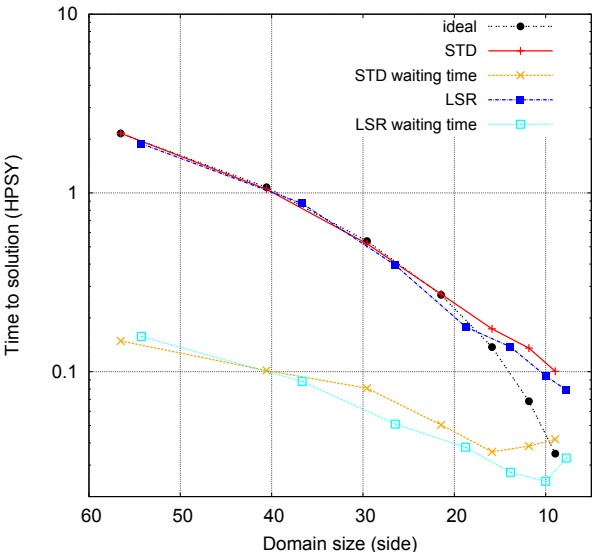

**Figure 1.** Time to solution (in hour per simulated year) as a function of the subdomain size (grid point number per subdomain side), BENCH configuration, same grid than global ocean at 1 degree resolution (ORCA1), including SI3 sea-ice subroutines, without any bathymetry (standard case, red line), or with land only subdomains removal (dark blue line). In each case, the time spent waiting boundary conditions is also plotted (resp. orange and light blue)

continents in the default usage of BENCH, prevent to test is the removal of subdomains entirely covered by land points
(Molines, 2004). Note that if we want to test a realistic bathymetry and the removal of land only subdomains in BENCH, we can use any NEMO configuration input file in BENCH (as in all NEMO configurations). In this case, we just have to define $ln\_read\_cfg = .true.$ and provide the configuration file name in the variable $cn\_domcf$ in the namelist block $namcfg$. A comparison of the BENCH scalability, without (named STD) or with land only subdomains removal (named LSR) was performed to assess the removal impact.

The subdomain size of configurations that remove or not land only subdomains, if decomposed with the same number of MPI processes, is different. A fair comparison of computing performance must be done for identical subdomain size. This is why the performance of Fig 1 is given as a function of the grid point number of a subdomain side (root mean square of the subdomain area), and not of the number of used resources, like in usual scalability plots. Performance is slightly better in LSR. The other information displayed in Fig 1, the simulation *waiting time*, represents the total elapsed time spent to wait MPI
communications for boundary condition update, and basically encompasses communication duration and computation load imbalance. The comparison of waiting time in STD and LSR helps to understand the origin of the small overall performance discrepancy. Most of it comes from LSR shorter waiting, which could be explained by the smaller amount of communication between processes, considering that in this configuration, some subdomains have boundaries with land only subdomains, thus no communication. However, scalability regimes, slopes and limits, in STD and LSR, for computations as for communications,





are practically the same. As already mentioned (Ticco et al., 2020), BENCH is able to reproduce computation performance of any usual NEMO configuration, and gives a simplified alternative for benchmarking work.

Since the 4.0 version release, the BENCH configuration was tested on various platforms and showed good numerical stability properties (Maisonnave and Masson, 2019). The stability of the BENCH configuration allows us to perform original benchmarks to easily test the potential impact of future optimization of MPI communications. BENCH can indeed run for quite a large number of time steps even if we artificially decide to skip MPI communications. We can, for example, figure out what would be the code scalability without any communications by simply adding a $RETURN$ command at the beginning of NEMO communication routine. If we add a $IF$ test with a modulo to decide if this $RETURN$ command is applied or not, one can have an idea of what would be the code performances if we could reduce the communications by a factor of N. By playing with the communication pattern in the BENCH configuration, one can therefore test the potential benefit of different ideas for the optimisation of the communications in NEMO before being really coding them.

### 2.3 Dedicated tool for communication cost measurement

The last piece of the puzzle that we implemented to set up an effective benchmarking environment in NEMO is to collect and summarizes information related to the performances of the MPI part in the code.

In NEMO, MPI communications are wrapped in a small number of high level routines co-located in a single Fortran file ($lib\_mpp.F90$). These routines provide functionalities for horizontal halo exchange and global averages or min/max operations between all horizontal subdomains. With very little code changes in this file, it is possible to identify and characterise the whole MPI communication pattern. This instrumentation does not replace external solutions, based on automatic instrumentation, e.g. with "Extrae-Paraver" (Prims et al., 2018) or "Intel Trace Analyzer and Collector" (Koldunov et al., 2019), which provide a comprehensive timeline of the exchanges. The amount of information collected by our solution is much smaller, and the possibilities of analysis reduced, but we are able to deliver without any external library, without additional computing cost or additional postprocessing, a simplified information for any kind of machine (portability).

A counter of the time spent in MPI routines ( i.e. "waiting time", as defined in subsection 2.2.3), is incremented at any call of a MPI send or receive operation in one hand, and at any MPI collective operations on the other hand. Symmetrically, a second counter is filled outside these MPI related operations. On each process, only two data are collected into two floats: the cumulative time spent sending/receiving/gathering or waiting MPI messages and the complementary period spent in other operations (named "computation time").

After a few time steps, we are able to produce in a dedicated output file called $communication\_report.txt$, the listing of the subroutines exchanging halos and how many times they did, on one hand, and the listing of the subroutines using collective communications on the other hand. The total number of exchanged halos, the number of 3D halos, and their maximum size are also provided.

At the simulation end, we also produce a separated counting, per MPI process, of the total duration of (i) halo exchanges for 2D/3D and simple/multiple arrays, (ii) collective communications needed to produce global sum/min/max, (iii) any other model operation independent from MPI (named "computing") and (iv) the whole simulation. These numbers exclude the first





and last time steps, so that any possible initialization or finalization operations were excluded of the counting. This analysis is
output jointly to the existing information related to the per-subroutine timing ($timing.output$ file).

## 3   Reducing or removing unnecessary MPI communications

This third section presents the code optimizations that were done following the implementation of the benchmarking environment described in the second section.

In a former study (Maisonnave and Masson, 2019), we relied on the measurement tool (section 2.3) to assess the performance
of the BENCH configuration (section 2.2). In this work, they focused only on global configurations with grid sizes equivalent to $1°$ to $1/12°$ horizontal resolution, including or not sea-ice and bio-geo-chemistry modules. Our analysis revealed that the global configurations scalability was limited by a major load imbalance due to the special halo exchange required by the north fold treatment. A fine study of the north fold communications pattern revealed that it was possible to further reduce the number of array lines involved in these specific communications, achieving a speed up of x1.4 at a new scalability limit.

Maisonnave and Masson (2019) also modified various NEMO subroutines to reduce (or group) MPI exchanges, like communicating a 3D halo in a single step instead of a sequence of 2D halos, or avoiding collective communication (i) during the prognostic variable sanity check, (ii) during the computation of a convergence criterion in sea-ice thermodynamics, and (iii) during bio-geo-chemistry tracer advection to spread negative value locally.

In this section, we propose to complement the work of Maisonnave and Masson (2019) by improving NEMO scalability
in regional configurations instead of global configurations. This implies to configure the BENCH namelist in a regional setup including open boundaries, as detailed in the annex A. We also deactivated the sea-ice component which was deeply rewritten in NEMO version 4 and necessitates a dedicated work on its performance.

As evidenced by Tintó et al. (2019), the MPI efficiency in NEMO is not limited by the volume of data to be transferred between processes but by the number of communications itself. Regrouping the communications that cannot be removed is
therefore a good strategy to improve NEMO performances. Our goal is thus to find the parts of the code which do the most of communications and next figure out how we can reduce this number either by removing communications that appear to be useless or by regrouping as much as possible the indispensable communications. Note that following the development of Tintó et al. (2019) a Fortran generic interface has been added to NEMO and makes the grouping of multiple communications extremely easy for the user.

In the configuration we are testing (BENCH with no periodicity, open boundaries and no sea-ice), almost 90% of the number of communications are due to the surface pressure gradient computation (44%) and the open boundary conditions (45%). The following optimizations will therefore focus on those two parts of the code. Note that in both cases, the involved communications transfer a very limited volume of data (from a single scalar to a 1-dimension array) which justifies even more the strategy proposed by Tintó et al. (2019).





## 3.1 Free Surface Computation Optimization


In most of the configurations based on NEMO, including in our BENCH test, the surface pressure gradient term in the prognostic ocean dynamics equation is computed using a "Forward-Backward" time-splitting formulation (Shchepetkin and McWilliams, 2005). At each time step $n$ of the model, a simplified $2D$ dynamic is resolved at a much smaller sub time step $\Delta t_*$ resulting in $m$ sub time step per time step, with $m$ ranging from 1 to $M (\approx 50)$. This $2D$ dynamic will then be averaged to

obtain the surface pressure gradient term.

In the previous NEMO version, each sub-time step completes the following computations :

$$
\begin{cases}
\eta^{m+\frac{1}{2}} = (\frac{3}{2}+\beta)\eta^m - (\frac{1}{2}+2\beta)\eta^{m-1} + \beta\eta^{m-2} \\
\overline{U_h}^{m+\frac{1}{2}} = (\frac{3}{2}+\beta)\overline{U_h}^m - (\frac{1}{2}+2\beta)\overline{U_h}^{m-1} + \beta\overline{U_h}^{m-2} \\
\tilde{U_h}^{m+\frac{1}{2}} = D^{m+\frac{1}{2}}\overline{U_h}^{m+\frac{1}{2}}\Delta e \\
Communication\ 1 \quad \text{on } \tilde{U_h}^{m+\frac{1}{2}} \\
\eta^{m+1} = \eta^m - \Delta t_*[div(\tilde{U_h}^{m+\frac{1}{2}}) + P - E] \\
Communication\ 2 \quad \text{on } \eta^{m+1} \\
\eta' = \delta\eta^{m+1} + (1-\delta-\gamma-\epsilon)\eta^m + \gamma\eta^{m-1} + \epsilon\eta^{m-2} \\
\overline{U_h}^{m+1} = \frac{1}{D^{m+1}}[D^m\overline{U_h}^m + \Delta t_*((1-r)g\ grad_x(\eta') - D^{m+\frac{1}{2}}fk \times \overline{U_h}^{m+\frac{1}{2}} + \overline{G})] \\
Communication\ 3 \quad \text{on } \overline{U_h}^{m+1}
\end{cases}
$$

With $\eta$ the sea surface height, $\overline{U_h}$ the speed integrated over the vertical, $\tilde{U_h}$ the flux, $D^m = H + \eta^m$ the height of the water column, $\Delta e$ the length of the cell, $P$ precipitation, $E$ evaporation, $g$ the gravity acceleration, $f$ the Coriolis frequency, $k$ the

vertical unit vector, $G$ a forcing term, $\beta, r, \delta, \gamma$ and $\epsilon$ are constants

NEMO uses a staggered Arakawa C grid, meaning that, among other, zonal velocities are evaluated at the middle of the eastern grid edges, meridional velocities at the middle of the northern grid edges and sea surface height at the grid center. Due to this feature, spatial interpolations are sometimes needed to get variables on other points that they are initially defined. For instance, $\eta^{m+\frac{1}{2}}$ must be interpolated to $U$ and $V$ points to be used in the equation $\tilde{U_h}^{m+\frac{1}{2}} = D^{m+\frac{1}{2}}\overline{U_h}^{m+\frac{1}{2}}\Delta e = (H +$

$\eta^{m+\frac{1}{2}})\overline{U_h}^{m+\frac{1}{2}}\Delta e$. Since adjacent points on both sides of the grid cells are needed in the interpolation, $\eta^{m+\frac{1}{2}}$ can not be directly interpolated on eastern $U$ points ghost cells ( grey arrows in figure 2) as there is no $T$ points on the east of an eastern $U$ point ghost cell in the MPI subdomain. For a similar reason, $\eta^{m+\frac{1}{2}}$ can not be interpolated on northern $V$ points ghost cells. It entails that $\tilde{U_h}^{m+\frac{1}{2}}$ is not directly computed on eastern $U$ point and northern $V$ points ghost cells and Communication 1 is used to update $\tilde{U_h}^{m+\frac{1}{2}}$ on those ghost cells.

The computation of $div(\tilde{U_h}^{m+\frac{1}{2}})$ defined at $T$ points requires values of $\tilde{U_h}^{m+\frac{1}{2}}$ on adjacent $U$ and $V$ points and therefore can not be completed on western and and southern ghost cells, Communication 2 updates the field on those cells. Similarly $grad_x(\eta')$ can not be computed over the whole MPI subdomain hence the Communication 3.



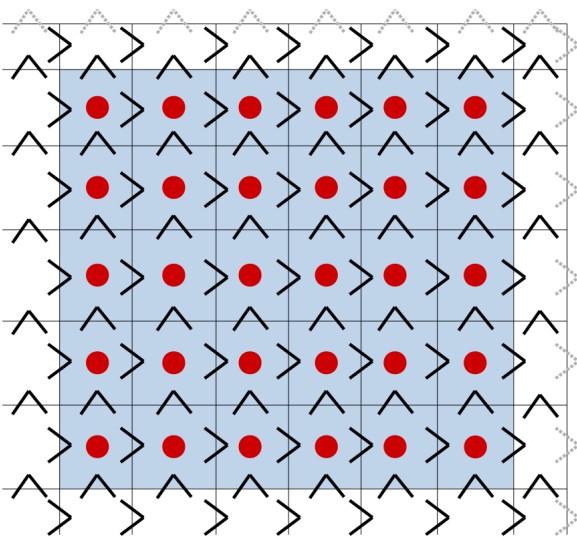

**Figure 2.** The MPI subdomain is bounded by the grid, the interior of the MPI subdomain is highlighted in blue while the ghost cells are in white. Black arrows show U and V points where $\eta^{m+\frac{1}{2}}$ (and therefore $\tilde{U_h}^{m+\frac{1}{2}}$) can be directly computed and grey arrows points where they can not be computed without communication. Red dots show T points where $div(\tilde{U_h}^{m+\frac{1}{2}})$ can be directly computed.

A careful examination of this algorithm is however showing that this communication sequence can be improved. As detailed on Figure 2, the computation of $div(\tilde{U_h}^{m+\frac{1}{2}})$ (red dots) defined at T points only demands correct values at the four adjacent U and V points (black arrows). Communication 1 on $\tilde{U_h}^{m+\frac{1}{2}}$ can be delayed and grouped together with Communication 2 on $\eta^{m+1}$ as values of $\eta^{m+\frac{1}{2}}$ on U and V points of northern and eastern ghost cells are not needed in the computation of $div(\tilde{U_h}^{m+\frac{1}{2}})$ on the interior of the MPI subdomain. Note that the communication on $\tilde{U_h}^{m+\frac{1}{2}}$ can not be removed altogether as the variable is also used for other purposes that are not detailed here.

Following this improvement, the number of communications per sub-time step in the time-splitting formulation has been reduced from 3 to 2. This translates in a reduction from 135 (44%) to 90 (29%) communications per time step in the surface pressure gradient routine for the examined configuration.

## 3.2 Open Boundaries Communication Optimisation

Configurations with open boundaries require to frequently correct fields on the boundaries. In the previous NEMO version, a communication must be carried out after the computation of boundary conditions to update values that are both on open boundary and on ghost cells. In the configuration we tested, with open boundaries and no sea-ice, 45% of the number of communications are due to open boundaries.

Boundary conditions in NEMO are often based on the Neumann condition, $\frac{\partial \phi}{\partial n} = 0$ where $\phi$ is the field on which the condition is applied and $n$ the outgoing normal, or the Sommerfeld condition $\frac{\partial \phi}{\partial t} + c\frac{\partial \phi}{\partial n} = 0$ where $c$ is the speed of the





transport through the boundary. For both boundary conditions the only spatial derivative involved is $\frac{\partial \phi}{\partial n}$. The main focus is therefore to find the best way to compute $\frac{\partial \phi}{\partial n}$ in various cases.

NEMO allows two kinds of open boundaries : straight open boundaries and unstructured open boundaries. As straight open boundaries along domain edges are far more common and easier to address, we will examine them first.

### 3.2.1   Straight Open Boundaries along domain edges

Figure 3.a shows schematic representation of the BENCH configuration with use here with with straight open boundaries on each sides that will be used to explain the optimization performed in this part of the code. The code structure of NEMO requires domains to be bordered by land points (in brown) on all directions except when cyclic conditions are applied. The four open boundaries (red stripes on each side of the domain) are thus located next to the the land points, on the second an before last rows and columns of the global domain.

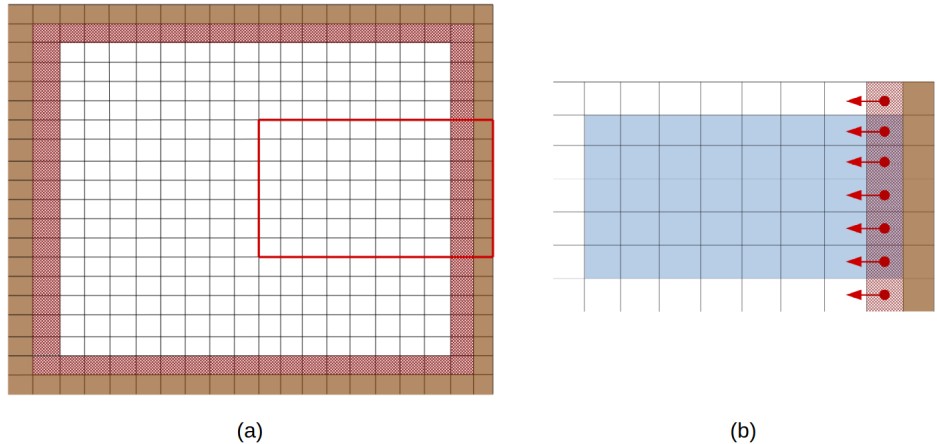

(a)                                                                 (b)

**Figure 3.** On the left, a configuration with straight open boundaries, the red line delimits a possible MPI subdomain detailed on the right. Brown cells are land cells and red hatched cells are open boundary cells. On the right, inner domain cells are in blue, red dots mark T points constituting the open boundary $((x_i, y_i))$ and red arrows the points orthogonal to the boundary $((x_{i-1}, y_i))$ used in the computing of $\frac{\partial \phi}{\partial n}\big|_{(x,y)=(x_i,y_i)}$.

Let us consider an MPI subdomain located on the eastern side of the global domain, represented by the red square on Figure 3.a and detailed on 3.b. Originally, the treatment of the open boundaries was performed only in the inner domain (red stripes over blue cells) and a communication phase was used to update the value of the open boundaries points located on the ghost cells (red stripes over white cells). In case of a straight longitudinal boundary with the exterior of the computational domain on the East, $\frac{\partial \phi}{\partial n}\big|_{(x,y)}$ will be discretized at a point $(x_i, y_i)$ by $\frac{\phi(x_{i-1}, y_i) - \phi(x_i, y_i)}{\Delta_{x_{i,i-1}}}$ where $\Delta_{x_{i,i-1}}$ is the distance between $x_i$ and $x_{i-1}$. When such conditions are applied, only values defined at points orthogonal to the open boundary are needed, yet such points are inside the MPI subdomain, even when $(x_i, y_i)$ is on a ghost cell (red stripes over white cells on Figure 3.b). Moreover, in the code, fields are always updated on ghost cells before open boundary conditions are applied, hence the entire



field on the MPI subdomain is properly defined and can be used. The computation of the boundary condition is thus possible over the whole boundary including the on ghost cells and there is no need for any communication update. When straight open boundaries along domain edges are used, this optimization gets rid of all the communications linked with open boundaries

computation.

### 3.2.2 Unstructured Open Boundaries

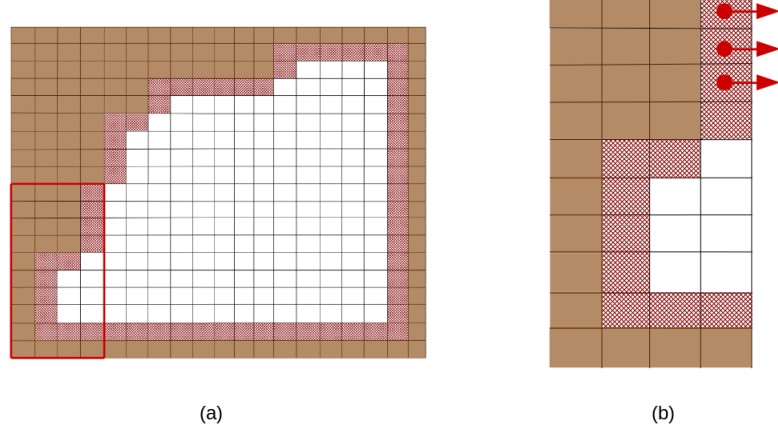

(a)                                                                 (b)

**Figure 4.** On the left, a configuration with straight open boundaries on the eastern and southern sides and unstructured open boundaries on the western and northern sides, the red line delimits a possible MPI subdomain detailed on the right. Brown cells are land cells and red hatched cells are open boundary cells. Red dots mark some T points of the open boundary ($(x_i, y_i)$) and red arrows the points orthogonal to the boundary used in the computing of $\frac{\partial \phi}{\partial n}|_{(x,y)=(x_i,y_i)}$.

Unstructured open boundaries allow the user to define any boundary shape. Figure 4.a shows an example of such an open boundary where open boundaries are defined all along land points. Depending on the MPI decomposition, open boundary cells can end up on ghost cells and facing the outside of the MPI subdomain, hence rendering direct computation of the boundary

condition $\frac{\partial \phi}{\partial n}|_{(x,y)=(x_i,y_i)}$ on some points ($(x_i, y_i)$) highlighted by a red arrow would require the value of $\phi$ on a point ($(x_{i+1}, y_i)$) outside of the MPI subdomain. Note that straight open boundaries can potentially be defined anywhere in the domain (and not only along the domain edges). In this case it is also possible that a straight open Boundary would be just tangential to the MPI domain decomposition. In such rare case, applying the open boundary conditions would also require a MPI communication. These rare

cases are in fact treated in the same manner as for the unstructured open boundaries and are thus automatically included in the procedure we implemented for unstructured open boundaries.

We chose to detect points where direct computation (i.e. without a communication phase) will be impossible during the initialization of the model and trigger communications only for MPI subdomains where at least one such point is present. This allows for the previous optimization to be compatible with unstructured open boundaries. Corner points of an unstructured





open boundary require also a specific attention when tracking the open boundary points that could need a communication phase. Indeed, on an outside corner, several points can be considered orthogonal to the open boundary. The choice of the neighbouring points involved in this computation will tell us if the treatment of the corner requires a MPI communication or not. When reviewing the corner points treatment in the previous NEMO version, we realized that the method chosen in some cases did not ensure symmetry properties (a reflection symmetry could change the results). We therefore decided to first

correct this problem in the physical application of the Neumann condition of corner points before finding and listing those which require a MPI communication. This first step is detailed in the next paragraph even if, formally, this is not a performance optimisation.

Applying the Neumann condition, $\frac{\partial \phi}{\partial n} = 0$, to an open boundary point equates to setting that point to the value of one (or the average value of several) of its neighbours that are orthogonal to the open boundary. The selected method must have reflection

and rotation symmetry properties and allow the open boundary point to be set to the most realistic value possible. The method used is illustrated on Figure 5 where contour lines of a field $\phi$ are in blue with the $\phi = 0$ contour line passing through the T point of an open boundary cell (red dot) on an outside corner of the open boundary. Contour lines are straight and the field is increasing linearly in diagonal or orthogonal directions. Red arrows indicate the best choice for the Neumann condition. In figure 5 the best choice is to take the average of the values of the closest available points. Here, applying the Neumann

condition is done by setting $\phi(x_i, y_i)$ to $\frac{\phi(x_{i+1}, y_i) + \phi(x_i, y_{i-1})}{2}$. Indeed, if only one of those two points was used, there would not be good symmetry properties and if the top right point was also taken into the average, the result would be a bit better in case 1 for a non linear field but worse in cases 2, 3 and 4.

Using the same method, we finally summarized all Neumann conditions and their neighbour contributions in 5 cases showed in Figure 6. All other possible dispositions are only rotations of one of these 5 cases. Thanks to this classification, we have

next been able to figure out, from the model initialisation phase, the communication pattern required by the treatment of the Neumann conditions and therefore restrict the number of communication phases to their strict minimum according to the chosen MPI subdomain decomposition.

### 3.3   Performance improvement

The computing performance is estimated using a realistic simulation of the West Atlantic between 100°West and 31°West and

8°South and 31°North. It includes straight open boundaries on the North, East and South, and continent. The simulation is to the $32^{th}$ of a degree with $2188 \times 1300 = 2\ 844\ 400$ mesh cells and 75 vertical levels. To reduce the impact of the file system on the measurements, the simulation does not produce any output.

In this test case, which is representative of the very large majority of uses, the open boundaries are straight and located along the edge of the domain. The communications related to open boundaries have therefore been completely removed and

the communications related to the surface pressure gradient are diminished by a third. As a result, in this configuration, the total number of communications per time step has been reduced by about 60% (Figure 7).

The number of core used is in line with the optimum dynamic sub-domain decomposition described in section 2.1. For each core number, 3 simulations of 1080 time steps were conducted and the computing time was retrieved at each time step of the



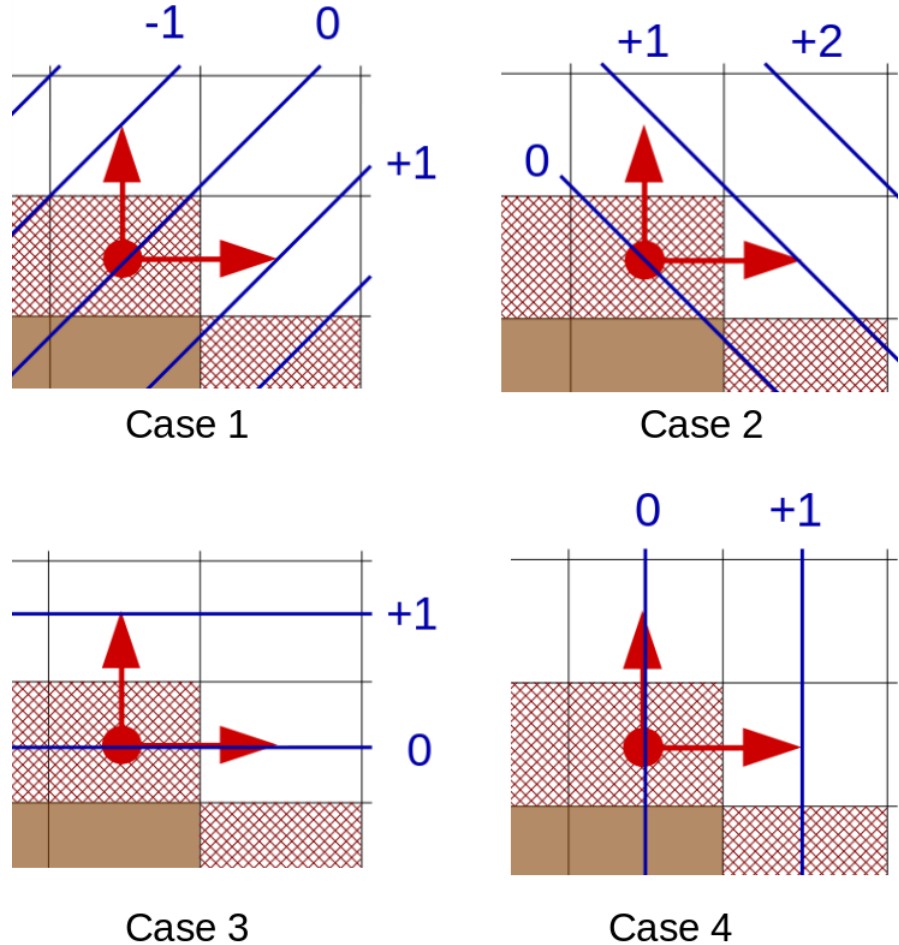

**Figure 5.** Brown cells are land cells and red hatched cells are open boundary cells. Red dots mark points from the open boundary and red arrows the best points to use to compute the Neumann condition, blue lines are contour lines of $\phi$ with the value of the contour line highlighted in blue.

model. Figure 8 shows that there are many outliers in the computing time that shift the mean to higher values while the median

is not sensible to it. As the exact same instructions are run at every time step, the outliers are likely to be a consequence of instabilities of the supercomputer or preemption. Since the median effectively filters out outliers it corresponds to the computing time one would get on a perfectly stable supercomputer with no preemption.

Figure 9 shows the improvement of NEMO strong scaling brought by our optimizations. In this figure, the model performance is quantified by the means of the number of simulated years that can be simulated during a 24-hour period (Simulated

Years Per Day, SYPD). For small numbers of core, the optimizations have no noticeable effect as the time spent in the communication phases is, in any case, very small. However, as the number of cores rises, each MPI subdomain gets smaller, the



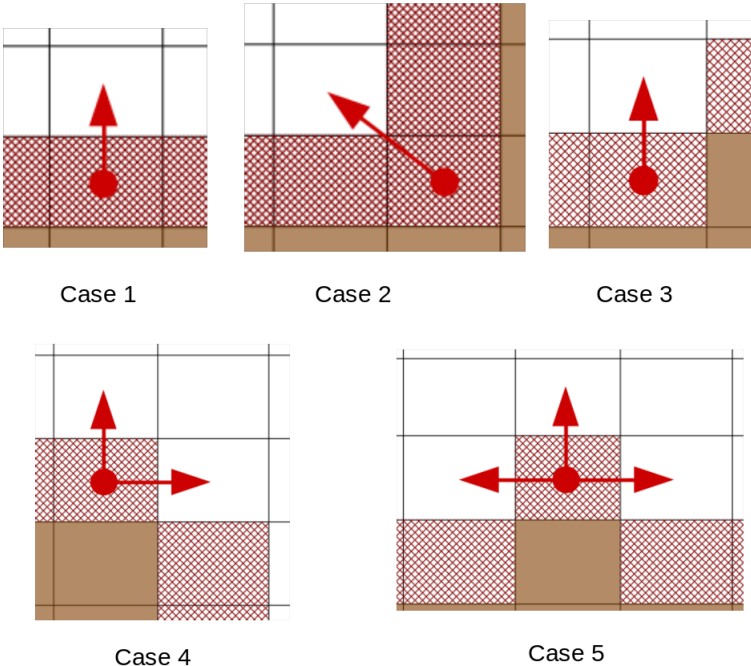

**Figure 6.** Brown cells are land cells and red hatched cells are open boundary cells. Red dots mark points from the open boundary and red arrows the points used in the computing of $\frac{\partial \phi}{\partial n}\big|_{(x,y)=(x_i,y_i)}$ for the Neumann condition.

communication load gets predominant and the optimizations bring clear improvements: the number or simulated years per day is higher in the new and optimized version of the code (orange curves are above the corresponding blue curves). The scalability curves built by considering the 1080 time steps (solid lines) are nevertheless quite noisy and the improvement is not as good as

expected: $20\%$ at best with even a negative value around $30,000$ cores.

     Filtering out the outliers by using the median gives results that are better and more robust. The scalability curves (dashed lines) are less noisy and, if we except the last point, the distance between the two dashed lines is steadily increasing as we use more cores. The impact of our optimizations is therefore more important at high numbers of core. The number of SYPD is, for example, increased by $35\%$ for $37,600$ cores when using the optimized version. One can also note that the optimized version

ran on $23,000$ cores simulates the same number of SYPD than the old version on $37,600$ cores which represents reduction of resources usage by nearly $40\%$.

     The differences between the unfiltered and the filtered results can be understood as a consequence that each core has similar chances of suffering from instabilities or going into preemption, hence, at high numbers of core, unusually slow cores are more common. Since communications tend to synchronise cores, a single slow core slows down the whole run. The median gets

rid of time steps in which preemption or great instabilities occur. Its indicates the performances we could get on a "perfect" machine which would not present any "anomalies" during the execution of the code. This machine is unfortunately not existing and the trend in the new machines architecture with always more complexity and heterogeneity suggests that performances



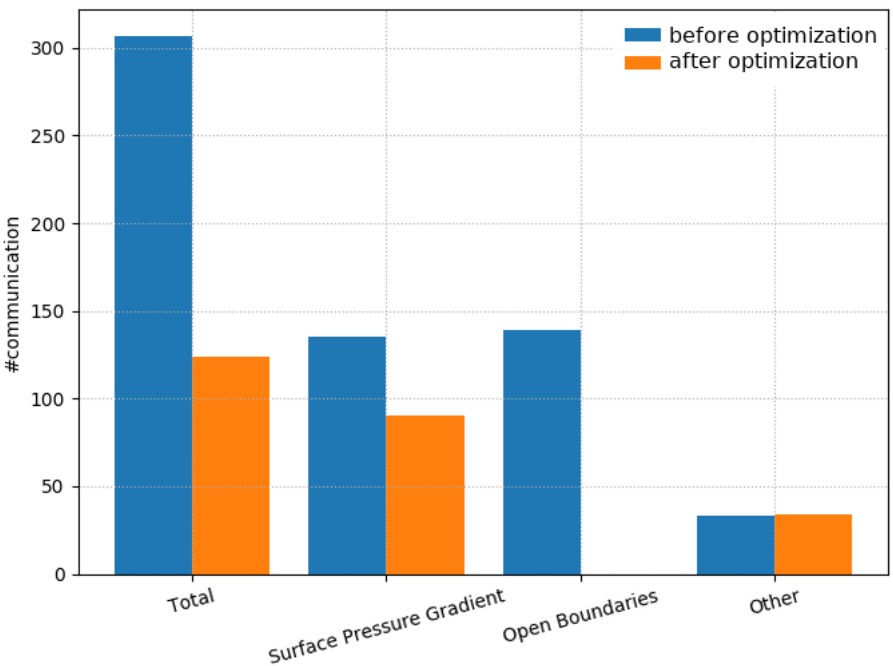

**Figure 7.** Comparison between the number of communication needed at each time step in NEMO before (blue) and after the optimizations of this section were carried out (orange) for a configuration with open boundaries and without ice.

"anomalies" during the model integration may occur more and more often to become a common feature. Our results point out this new constrain which is already already eroding a significant part of our optimisation gains (from $40\%$ to $20\%$) and will require to be taken into account in the future optimizations of the code.

## 4   Conclusions

We presented in this paper the new HPC optimisations what have been implemented in NEMO 4.0, the current reference version of the code. The different skills we gathered among the co-authors allowed us to improve NEMO performances while facilitating its use. The automatic and optimized domain decomposition is a key feature to perform a proper benchmarking work but it also benefits to all users by selecting the optimum use of the available resources. This new feature also points out possible waste of resources to the users, making them aware of the critical impact of the choice of the domain decomposition on the code performance. The new BENCH test case results from a close collaboration between ocean physicist and HPC engineers. We distorted the model input, boundary and forcing conditions in such way that the model is stable enough to do benchmark simulations in any configuration with the less possible input files as possible (basically the namelist files). Note that the stability of this configuration is even allowing developers to carry out some unorthodox HPC tests as, for example,





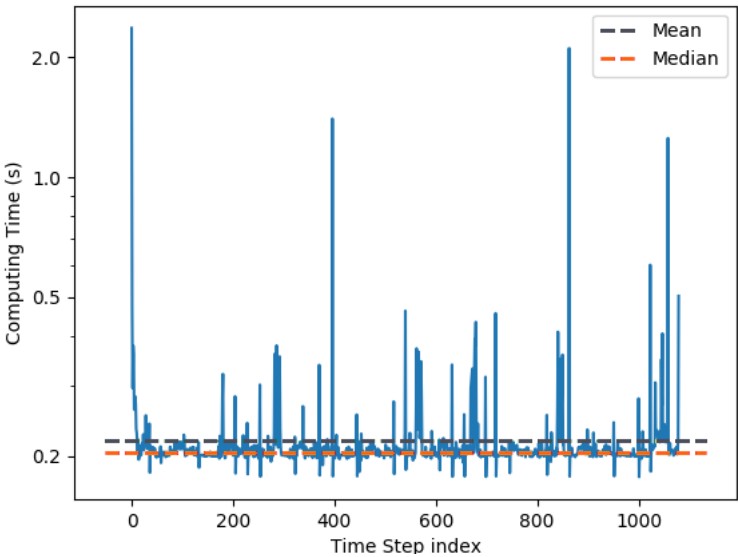

**Figure 8.** Computing time (in seconds) per time step for a North-Atlantic simulation run with 2049 cores. The grey (resp. orange) dashed line shows the mean (resp. median) computing time of one time step.

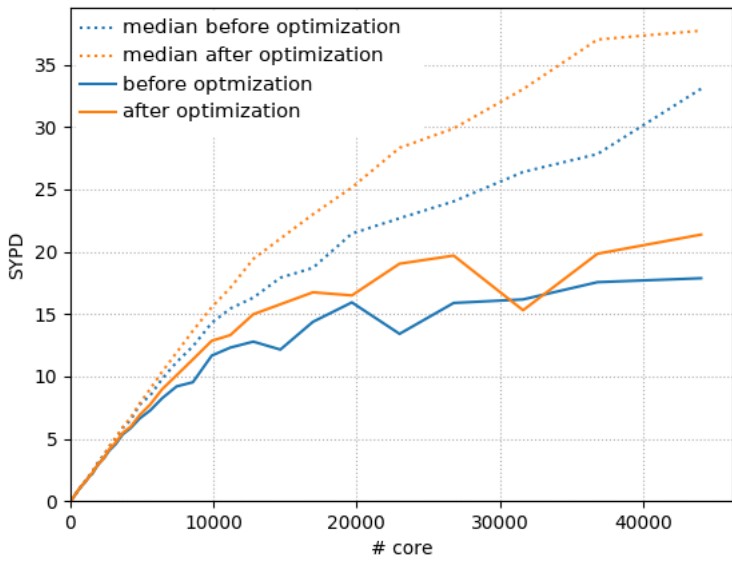

**Figure 9.** Strong scaling performances: Simulated Years Per Day (SYPD) as a function of the number of core used in the simulation

artificially suppressing a part of the MPI communications to test the potential benefit of further optimisations before coding them.





In this 4.0 release, code optimisation was targeting the scalability by reducing the number of communications. The present paper focuses on two parts of the code: (1) the computation of the surface pressure gradient, which does about 150 communications per model time step and (2) the treatment of the open boundaries conditions that are also doing a similar amount of communications per model time step. We managed to reduce the communications by 30% in the routine computing the surface pressure gradient. The results are even more spectacular in the part of the code dealing with the open boundaries as we managed to suppress all communications in the very large majority of cases. Note that this optimisation work gave us also the opportunity to improve the algorithm used in the treatment of some unstructured open boundaries with a tricky geometry. Several conclusions can be drawn from the analysis of the performance improvements obtained with this very large reduction (from $\sim 300$ to $\sim 125$, that is to say $\sim 60\%$) of the total number of communications per model time step. First, as expected, this optimization has an impact only once we use enough cores (>500 in our case) so the communications have a significant weight in the total elapsed time of the simulation. Second, the elapsed time required to perform one model time step is far from being constant as it should be in theory. Some time steps are much longer to compute than the median time step. This significant spread in the model time step duration requires that we use the median instead of the mean value when comparing performances of different simulations.

These results are suggesting further investigations for future optimizations. The large variability of the elapsed time needed for one time step suggests that the performance of the machines are definitely not constant. They are, in fact, varying in time and between the cores in a manner that is much larger than what we originally expected. This behaviour can be explained by many things (preemption, network load, etc...) and benchmarks on different machines are suggesting that this heterogeneity in the functioning of the supercomputers will happen more and more as we are using more and more cores. Code optimisations will have to take this new constrain into account. We will have to adapt our codes in order to absorb (or at least limit the effects) of the asynchrony that will appear during the execution on the different cores at a growing frequency. The way we perform the communication phase in NEMO, with its point-to-point synchronisation between, first, east-west neighbours and, next, north-south neighbours will have to be revisited for example with non-blocking send and receive or with new features such as neighbour collective communications. In recent architecture, the number of cores inside a node has increased. This leads to a two-level parallelism where the communication speed/latency differs for inter-node exchanges and intra-node exchanges. Inter-nodes communications are probably slower and are probably the source of more asynchrony than the inner-nodes communications. A possible optimisation could therefore be to minimize the ratio of inter-node versus inner-nodes communications. The figure 10 illustrates this idea. The domain decomposition of NEMO is represented with each square being an MPI processes and each node a rectangle of the same colour. In this representation, the perimeter of a "rectangle-node" directly gives the number of inter-node communications (if we say to simplify that the domain is bi-periodic and each MPI process has always 4 neighbours). If we make the hypothesis that that each node can host $x^2$ MPI processes, these $x^2$ processes are placed by default on a "line-node" with a perimeter of $2x^2 + 2$. In this case, the ratio of inter-node versus inner-nodes communications is $(2x^2 + 2)/4x^2$. Now, if we distribute the $x^2$ processes on a "square-node", the number of inter-node communications becomes $4x$ and the ratio $4x/4x^2 = 1/x$. If we take $x = 8$ for 64-core nodes, we get a reduction of -75% of the number of inter-node communications when comparing the "line-node" dispatch (130) with the "square-node" dispatch (32). The ratio of inter-node



versus inner-nodes communications drops from 50% to 12.5%. These numbers are of course given for ideal cases where the number cores per node is a square. We should also consider additional constrains like the removal of MPI processes containing

only land points, the use of some of the cores per node for XIOS, the IO server used and NEMO. The optimal dispatch of the MPI processes in a real application is maybe not so trivial but there is certainly here an easy way to minimize the inter-node communications, which could be an advantage when we will be confronted with the occurrence of more and more asynchrony.

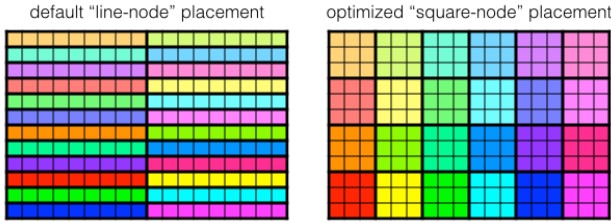

**Figure 10.** NEMO domain decomposition with the dispatch of the MPI processes among the different nodes. In this schematic representation, each square represents one MPI process of NEMO. We consider that NEMO is distributed over 216 MPI processes and that each node has 9 cores. The distribution of the MPI processes on the cores is represented by their colour: processes on the same node have the same colour. left panel: default dispatch of the MPI processes in line, right panel: optimized dispatch of the MPI processes in square

*Code availability.* The NEMO source code and all the code developments described in this paper are freely available on NEMO svn depository at the following address: https://forge.ipsl.jussieu.fr/nemo/browser/NEMO/releases/r4.0

or alternatively here : https://doi.org/10.5281/zenodo.5566313

*Acknowledgements.* This research has been supported by the European Commission, H2020 European Data Infrastructure (ESiWACE (grant no. 675191)). The authors acknowledge the Météo-France, TGCC and IDRIS supercomputer administration for having facilitated the scalability tests.

## Appendix A: Namelist configuration to use BENCH with open boundaries

The BENCH configuration, see section 2.2, was used by Maisonnave and Masson (2019) in its global configuration including the east-west periodicity and the north pole folding. As detailed in section 2.2.2, the BENCH configuration can be adapted to any purpose through the use of the configuration file: namelist_cfg.

This annex shows the parameters to be modified or added in namelist_cfg in order to use the BENCH configuration with strait open boundaries used in section 3.

```
!-------------------------------------------------------------------
&namusr_def    !   User defined :   BENCH configuration
```





```
     !-------------------------------------------------------------------
        nn_perio   =   0         ! no periodicity
     /


     !-------------------------------------------------------------------
     &nambdy          !  unstructured open boundaries
     !-------------------------------------------------------------------
        ln_bdy          = .true.    !  Use unstructured open boundaries
nb_bdy          = 4         !  number of open boundary sets
        ln_coords_file = .false.,.false.,.false.,.false.
        cn_dyn2d       = 'flather','flather','flather','flather'
        nn_dyn2d_dta  = 0,0,0,0
        cn_dyn3d       = 'frs','frs','frs','frs'
nn_dyn3d_dta  = 0,0,0,0
        cn_tra          = 'frs','frs','frs','frs'
        nn_tra_dta     = 0,0,0,0
        cn_ice          = 'frs','frs','frs','frs'
        nn_ice_dta     = 0,0,0,0
nn_rimwidth   = 5,5,5,5
     /
     !-------------------------------------------------------------------
     &nambdy_index
     !-------------------------------------------------------------------
ctypebdy = 'E'
        nbdyind  = -1
     /
     !-------------------------------------------------------------------
     &nambdy_index
!-------------------------------------------------------------------
        ctypebdy = 'W'
        nbdyind  = -1
     /
     !-------------------------------------------------------------------
&nambdy_index
     !-------------------------------------------------------------------
        ctypebdy = 'N'
        nbdyind  = -1
     /
!-------------------------------------------------------------------
     &nambdy_index
     !-------------------------------------------------------------------
        ctypebdy = 'S'
        nbdyind  = -1
540  /
```





*Author contributions.* Sebastien Masson and Erwan Raffin supervised this work. Sebastien Masson developed the optimization of the sub-domain decomposition. The BENCH configuration was introduced by Sebastien Masson and Eric Maisonnave. The reduction of MPI communications and the subsequent analysis was executed by Gaston Irrmann and Sebastien Masson. David Guibert implemented the reduction of inter node communication. All the authors have contributed to the writing of this paper.

*Competing interests.* The authors declare that they have no conflict of interest.



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
