# Peer review of "Improving Ocean Modelling Software NEMO 4.0 benchmarking and communication efficiency"

_Geoscientific Model Development, 2021_

## Author Response (AR1)

**Author's response**

January 6, 2022

**1 Discussion with reviewer 1**

**Comment from reviewer 1**

Improving the computational efficiency of open-source community ocean models, such as NEMO v4.0, is of great benefits to climate modelling and operational ocean forecasting. This manuscript reports recent contributions through the collaboration of numerical modelling and HPC experts in several aspects: (1) the creation of a benchmark (BENCH) configuration for testing the model efficiency; (2) the automatic definition of "optimal" domain decomposition; and (3) code modifications to reduce the communications among subdomains in two bottleneck routines. I believe the work described in this manuscript are valuable for both model developers and users.

In my review, I did not dig in all the technical details described in this manuscript because I am not an expert in parallel computing. For this, please consider this as a partial review. My focus is on the presentation aspect, as I find the manuscript is rather difficult to read due mainly to the writing style that makes the presentation neither concise nor accurate. There are many long sentences that are not easy to follow. There are also numerous grammar mistakes or typos throughout the manuscript. I will give some examples in the following "Specific comments" and "Technical corrections", but the list is far from being complete. I suggest that the manuscript needs a thorough revision/editing before being accepted for publication in a journal for broader readers.

**Response to reviewer 1**

Thank you for your input and corrections. We have taken them into account and revised the whole manuscript, changing formulations to improve conciseness and clearing ambiguities. We hope you will find this new version of our manuscript clearer and easier to read.

**2   Discussion with reviewer 2**

**Comment from reviewer 2**

The authors point out that the scalability of global configuration model is limited by the load imbalance due to the special halo exchange required for the north fold. This fact does not seem to be reflected in the design of the domain decomposition algorithm. Would it be appropriate to attempt to address the load imbalance at this stage. Does the proposed domain decomposition algorithm lead to improved load balance? It would be desirable to supplement the text with figure illustrating an example of the algorithm in action.

**Response to reviewer 2**

The domain decomposition, described in this paper, does not directly take into account the load increase due to the special halo exchange for the north pole. We have nevertheless introduced an optimization for this north pole issue that was not described in the original version of our manuscript. We added to the paper a new section (2.1.4), which details how we try to minimize the j-size of the processes along the northern edge of the domain. As requested by the reviewer, we also added a figure (number 1) to detail how this optimization applies to a one of NEMO global configurations. We chose ORCA025 as an example that is not too small and not too big.

**Comment from reviewer 2**

The effort to group communications togeather in one case or remove altogeather in the other translates to significant computational gains. Figure 8 shows a frequent occurance of abnormally slow time steps, with the factor of up to x10. Is it possible to verify that the slowdown occurs for random cores or perhaps is it systematic for a particular core. I would suggest to run the tests of a multiplicity of full electrical groups to avoid other jobs affect the network congestion in your experiments.

**Response to reviewer 2**

To investigate the issue regarding abnormally slow time steps we had to break down the timing much further. Indeed, the slowdown originates on a single core of the simulation at a very fine time scale. Because of the communications, by the end of the time step the delay produced by a slower core will be propagated to a significant portion of the simulation. We hence timed each call to any MPI routine inside NEMO's communication routine and the timed the execution in between MPI calls.

On the figure 1, on each plot the horizontal axis is the index of the call to the communication routine in the time step, on the vertical axis is the execution time on a logarithmic scale. The different figures correspond to : (top,

[Figure]

Figure 1: Execution time (s) of each phase of the communication pattern.

left to right) the East-West communication, small computations following that communication, the North-South communication, (bottom, left to right) small computations and deallocation following the North-South communication, the computations following the whole communication (the code outside the communication routine) and the total time spent in the communication routine. For each figure the percentiles computed over the course of a simulation of 2080 time steps are plotted in colors and the average in bold black. In this figure, the measurement is done on the $39^{th}$ allocated core (MPI rank), the positions of the core is highlighted in the top middle graph. This core was chosen at random but all the cores of the simulation yield very similar results.

East-West communications are intra-node and do not involve interconnect network. As even for intra-node communications the maximum of the execution

time is very high, the location of the nodes even on distant parts of the machine is not the cause of the slowdowns. The cores responsible for the slowdowns are also seemingly random as each core seems to behave in a similar way. As the slowdown occurs on random cores it is likely to be caused by the hardware or OS interference. We have tried on several supercomputers and still found this issue, occurring more often on some than others. We are currently investigating ways to make NEMO more resilient to such slowdowns.

The article was revised very concisely by adding the following sentence to the part about outliers : "A finer analysis showed that the slowdowns occur on random cores of the simulation."